# MAX-SPEEDUP SPECULATIVE SAMPLING: A GENERIC TREE CONSTRUCTION PRINCIPLE

## ABSTRACT

Speculative sampling has emerged as a promising approach for accelerating large language models (LLMs) inference by leveraging a lightweight draft model to propose multiple candidate tokens, which are then verified in parallel by a target model. Recent methods enhance this process by structuring candidate sequences into a token tree for more efficient verification. However, existing tree construction methods rely excessively on acceptance length as a proxy for speedup. This indirect pursuit renders it challenging to achieve the optimal tree structure for maximum speedup. In this paper, we first revisit prior approaches and find that they suffer from two key limitations: analytical intractability and the assumption of node homogeneity. We then redefine the costs and benefits of each tree node, derive a function that characterizes the relationship between time reduction and draft length, and prove its convexity. Finally, we extend this analytical framework to tree structures and propose a general principle for tree construction aimed at maximizing speedup. Applying this principle to state-of-the-art tree-based speculative sampling methods consistently yields significant gains, improving overall performance by 4% to 14% and achieving end-to-end speedups of 1.97× to 2.68×. The implementation is publicly available at: https://anonymous.4open.science/r/GTCP-CC76/README.md.

## 1 INTRODUCTION

Large Language Models (LLMs) have exerted a transformative impact across a wide range of applications, from real-time human–computer interaction to complex reasoning tasks (Achiam et al., 2023; Chang et al., 2024; Radford et al., 2019; Touvron et al., 2023a;b; Chiang et al., 2023; Jiang et al., 2023). However, their reliance on autoregressive decoding as their core generative paradigm gives rise to substantial inference latency, stemming from its inherently sequential, token-by-token operational nature (Gante, 2023; Xiao et al., 2023; Huang et al., 2025). As model sizes and output lengths grow, this sequential decoding latency becomes a critical performance bottleneck, limiting deployment in real-time scenarios where low latency is essential.

Drawing inspiration from speculative execution in processor-level optimization (Gabbay & Mendelson, 1996; Smith, 1998; Burton, 2012; Hennessy & Patterson, 2011), speculative sampling has emerged as a promising technique for accelerating LLM inference (Gante, 2023; Xia et al., 2023; Leviathan et al., 2023; Stern et al., 2018; Chen et al., 2023). This approach leverages a draft model to efficiently generate multiple candidate tokens, which are subsequently validated in parallel by the target model. By reducing the number of autoregressive decoding steps, this approach yields significant speedups without compromising the quality of generated outputs.

Building on speculative sampling for LLM acceleration, recent research has introduced a tree-structured organization of candidate sequences, allowing target models to validate candidate token sequences in parallel and retain the longest valid acceptance trajectory (Miao et al., 2024; Spector & Re, 2023; Sun et al., 2024). Despite notable progress in tree-based speculative sampling, these approaches exhibit critical limitations that demand rigorous examination. As illustrated in Fig. 1(a), existing methods typically aim to construct trees with maximum average acceptance lengths under predetermined structural constraints. For instance, Sequoia (Chen et al., 2024b) proposes a tree construction method to maximize this metric, which utilizes dynamic programming algorithms under fixed constraints on the number of nodes, maximum depth, and maximum width. Similarly,

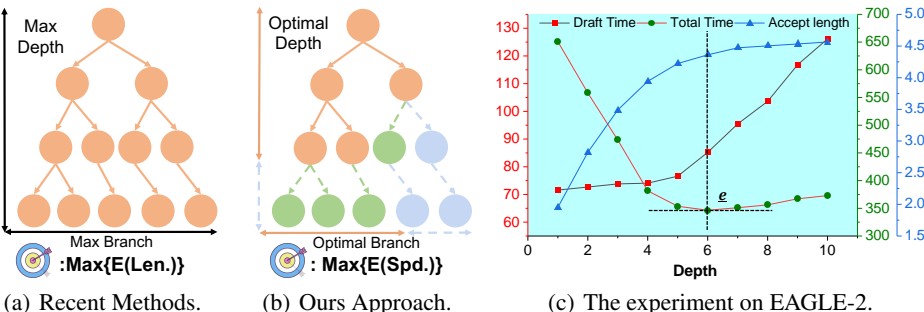

(a) Recent Methods.     (b) Ours Approach.     (c) The experiment on EAGLE-2.

Figure 1: (a) shows the variation of Draft time, Total time, and Acceptance Length with the increase in tree depth. (b) and (c) show the difference between recent methods and our method, where orange represents selected nodes, green nodes are positive benefits, and blue nodes are negative benefits. Our method automatically selects nodes based on benefits and costs.

EAGLE-2 (Li et al., 2024b) dynamically identifies a fixed number of nodes with the highest acceptance probability within these same constraints. While these methods prioritize greater average acceptance lengths, pursuing this metric does not inherently guarantee superior speedup. Specifically, this strategy entails undue time and resource expenditures when exploring low-reward nodes, fundamentally precluding the discovery of globally optimal verification trees. Thus, a critical gap remains in establishing a general principle for tree construction with maximum speedup.

To address this, we identify a fundamental trade-off: total inference time is governed by the duration of each decoding step and the number of iterations. Speculative sampling, which generates multiple tokens per verification, reduces the number of decoding steps—thereby accelerating the inference process. As shown in Figure 1(c), as tree size expands, the growth in acceptance rate and the reduction in decoding steps plateau (blue line), whereas drafting time escalates, increasing the duration of each decoding step (red line). When the advantage of fewer decoding steps is outweighed by the expense of prolonged drafting time, total inference time begins to climb (rto the right of the green line at point e). Overall, as the tree size increases, the total inference time first decreases and then increases (green line). This pattern stems from the trade-off between fewer decoding steps and a longer duration of each decoding step. Our analysis reveals that total time is well-approximated by a convex function of tree size, reaching a minimum at the optimal size (point e).

Building on this insight, we first revisit the formulations of Leviathan et al. (2023) to analyze the relationship between expected speedup and draft length. However, the prior approach is analytically intractable and relies on the unrealistic assumption of node homogeneity. Thus, we instead use the individual, non-identical acceptance rate of each node rather than the average acceptance rate to redefine its benefit and cost. Based on this refinement, we derive a new expression for the time reduction as a function of draft length and prove that it is convex. Applying the same reasoning, we extend the framework to tree construction and propose a general principle that achieves maximum speedup. According to this principle, selecting all nodes with positive benefits maximizes both time reduction and speedup, while the tree's structural properties—including node count, depth, and width—are automatically determined (see Figure 1(b), where green nodes with positive benefits are included and blue nodes with negative benefits are discarded). We validated this principle by integrating it into state-of-the-art static and dynamic tree construction methods, achieving 4–7% and 7–14% performance gains, with speedups of 1.97–2.19× and 2.54–2.68×, respectively. Additional experiments further confirm the principle's generality, stability, and memory-efficient acceleration.

In summary, our main contributions can be summarized as follows: 1) We identify the key trade-off in LLM tree-based speculative sampling, proving that total inference time is a convex function of tree size which grounds speedup-oriented construction. 2) We redefine the benefit-cost formula for each node and propose a general tree construction principle for maximizing speedup. 3) Experimental results demonstrate improved speedups across diverse model sizes and tasks, while ablation studies validate the proposed principle's adaptability to variations in batch sizes, sampling temperatures, and memory constraints.

## 2 BACKGROUND

### 2.1 SEQUENCE-BASED SPECULATIVE SAMPLING

Leviathan et al. (2023) analyzes the speculative sampling algorithm's performance by deriving expressions for the expected number of accepted tokens per iteration and the anticipated wall-clock speedup relative to standard autoregressive inference with only the target model. They define the average acceptance rate ($\alpha \in [0, 1]$) as the probability of accepting a token ($x_i$), assuming acceptance decisions are i.i.d. for simplicity. Under this assumption, the expected number of generated tokens is ($\frac{1-\alpha^{\gamma+1}}{1-\alpha}$). If $c$ denotes the ratio of inference time between the draft and target models, the expected speedup is given by ($\frac{1-\alpha^{\gamma+1}}{(1-\alpha)(\gamma c+1)}$).

### 2.2 TREE-BASED SPECULATIVE SAMPLING

Tree-based methods' core advantage is parallel evaluation of multiple speculative candidates to boost speculative performance. These candidates form a token tree, where each node represents a speculated token sequence, and this structure enables hierarchical exploration of generation paths. SpecInfer (Miao et al., 2024) pioneered this paradigm: it merges multiple candidate sequences into a token tree via prefix sharing, and uses a dedicated tree attention mask to let large language models (LLMs) verify the whole tree in parallel. Compared to sequential methods, this improves the throughput of parallel lexical verification while maintaining generation quality. Recent work optimizes token tree structure and size for computational efficiency: Sequoia (Chen et al., 2024b) uses a hardware-aware optimizer to select optimal trees, leveraging computational resources to maximize speedup; EAGLE-2 (Li et al., 2024b) adds runtime dynamic tree expansion (based on prediction confidence) and context-aware construction to improve acceptance rate. More details are in the Appendix A.2.

## 3 METHODOLOGY

We first analyze the relation between speedup and the structural properties of the verification tree (branch, depth, node count) in section 3.1 and get the optimal tree construction method. Then, we apply it to the advanced static and dynamic tree in section 3.2.

### 3.1 THE RELATION BETWEEN SPEEDUP AND TREE STRUCTURE

As mentioned in Section 2.1, Leviathan et al. (2023) derived the formula for expected speedup:

$$\mathbb{E}(\text{Spd.}) = \frac{1-\alpha^{\gamma+1}}{(1-\alpha)(\gamma c+1)}, \alpha \in (0, 1), c \in (0, 1). \tag{1}$$

Here, $\alpha$ is the average acceptance rate. The $\gamma$ denotes the drafting length of speculative sampling, and the cost coefficient $c$ is the ratio between the wall-clock time of a single run of the draft model $M_D$ and that of the target model $M_T$ ($c = \frac{M_D}{M_T}$).

From the above formula, we can derive: 1) When $\gamma = 0$, $\mathbb{E}(\text{Spd.}) = 1$; 2) When $\gamma = 1$, $\mathbb{E}(\text{Spd.}) = \frac{1+\alpha}{1+c}$; 3) When $\gamma \to +\infty$, $\mathbb{E}(\text{Spd.}) \to 0$. When $\alpha > c$, there exists a $\gamma$ such that $\mathbb{E}(\text{Spd.})$ achieves a value of at least $\frac{1+\alpha}{1+c}$. In practice, $\alpha$ is typically greater than $c$.

Additionally, as observed in Experiment 1(c) (and further illustrated in Figure 2), the expected speedup $\mathbb{E}(\text{Spd.})$, initially increases with $\gamma$, reaches a maximum, and subsequently declines. This trend raises a fundamental question: **How can we characterize this maximum?** However, directly analyzing the formula 1 to derive the relationship between $\gamma$ and $\mathbb{E}(\text{Spd.})$ is analytically challenging. Furthermore, its average acceptance rate $\alpha$ is based on the node homogeneity assumption (the acceptance rate of each

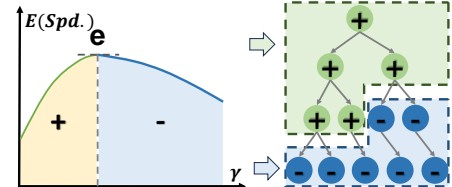

Figure 2: Nodes with costs and benefits.

node is the same), while the true acceptance rate of each node varies across different situations. For instance, the formula 1 models the probability of accepting a $\gamma$ length sequence simply as $\alpha^\gamma$. In reality, the true acceptance probability for a path is the product of the individual, non-identical token acceptance rates: $\alpha_\gamma^* = \prod_1^\gamma \alpha_i$, where $\alpha_1 \neq \alpha_2 \neq ... \neq \alpha_\gamma$. Therefore, we attempt to use the true global acceptance probability to redefine the benefit and cost of each node. By selectively including only nodes with a positive net benefit (i.e., where the benefit outweighs the cost), the speedup is maximized at point $e$ in Figure 2. This selection process naturally induces an optimal tree structure for verification.

**Sequence-based principle**    During speculative sampling, drafting one token needs $M_D$ wall-clock time and verifying $\gamma$ tokens need $M_T + \Phi_{M_T}(\gamma)$ wall-clock time. In order to draft $\gamma$ tokens, we define the addition cost of drafting the $i$ token as:

$$M_D + \Phi_{M_T}(\gamma), \quad \text{if } \gamma < K, \text{ then } \Phi_{M_T}(\gamma) = 0. \tag{2}$$

$\Phi_{M_T}(\gamma)$ is a function related to the additional time that the target model takes to verify $x$ tokens. It is usually equal to 0 up to a certain threshold $K$ (Varies on different devices). And if a drafted token is accepted, the system saves $M_T$ wall-clock time. Thus, we define the benefit as:

$$M_T \cdot (\hat{\alpha}_i + \epsilon) \tag{3}$$

Here, $\hat{\alpha}_i$ denotes the estimate of the acceptance probability for the $i$-th token, while $\epsilon$ represents the error between the true acceptance probability ($\alpha^*$) and its estimate ($\hat{\alpha}$). The formula for the time reduction in one step can be derived as:

$$\Delta T(\gamma) = \sum_0^\gamma \{M_T(\hat{\alpha}_i + \epsilon) - M_D\} = M_T \sum_0^\gamma \hat{\alpha}_i - M_D \cdot \gamma + \epsilon, \quad \text{where } \gamma < K. \tag{4}$$

Typically, $\hat{\alpha}_1$ is relatively high and speedup is typically achieved ($\Delta T(1) = M_T \cdot \alpha_1 - M_D + \epsilon > 0$). From the above formula, we can derive: **(1)** When $\gamma = 1$, $\Delta T(\gamma) > 0$; **(2)** When $\gamma \to +\infty$, $M_D \equiv$ const., $(M_T \cdot \hat{\alpha}_\gamma) \downarrow$; When $\gamma \in (1, \hat{\gamma})$, $M_D < M_T \cdot \hat{\alpha}_\gamma$, $\Delta T(\gamma) \uparrow$, and when $\gamma \in (\hat{\gamma}, +\infty)$, $M_D > M_T \cdot \hat{\alpha}_\gamma$, $\Delta T(\gamma) \downarrow$. Therefore, $\Delta T$ is a convex function that reaches its maximum at $\hat{\gamma}$, which is consistent with the experiment 1(c) and Figure 2. Furthermore, a positive-benefit node is defined as a node where the benefit is greater than to the cost:

$$M_T(\hat{\alpha}_i + \epsilon) - M_D > 0 \iff M_T(\hat{\alpha}_i - \frac{M_D}{M_T} + \epsilon) > 0, \quad M_T, M_D > 0. \tag{5}$$

Let $c = \frac{M_D}{M_T}$. By selecting all positive-benefit nodes (where $\hat{\alpha}_\gamma > (c - \epsilon)$), we can automatically determine $\hat{\gamma}$. Equation 5 is the sequence-based principle for determining the optimal draft length $\hat{\gamma}$. When assuming the node homogeneity, the estimated speedup $\mathbb{E}(\text{Spd.})$ of our method is consistent with that obtained by Leviathan et al. (2023). More details are in Appendix A.9.

**Tree-based principle**    The definition of tree-based methods is similar to that of sequence-based methods, except that tree methods have additional definitions: depth $D$, branch $B$, and node $N$. For a tree with $N$ nodes, the cost of a single node is independent of depth and branch, and is defined as:

$$M_D + \Phi_{M_T}(N), \quad \text{if } N < K, \text{ then } \Phi_{M_T}(N) = 0. \tag{6}$$

If the node at depth $d$ and branch $b$, its benefit is defined as: If the $d$-depth and $b$-branch node is selected, its benefit is similarly defined as:

$$M_T \cdot (\hat{\alpha}_{d,b} + \epsilon) \tag{7}$$

Here, $\hat{\alpha_{b,d}}$ represents the acceptance probability of the node at branch $b$ and depth $d$. And, the formula for the time reduction in one step can be derived as:

$$\Delta T(N) = \sum_0^N \{M_T(\hat{\alpha}_{d,b} + \epsilon) - M_D\}, \quad \text{where } N < K. \tag{8}$$

Then, the overall positive nodes must satisfy:

$$M_T(\hat{\alpha} + \epsilon) - M_D > 0 \iff M_T(\hat{\alpha}_{d,b} - c + \epsilon) > 0, \quad M_T, M_D > 0. \tag{9}$$

By selecting all nodes with positive benefits, we can automatically determine the depth $\hat{D} = \max_d\{\hat{\alpha}_{d,b} > (c - \epsilon)\}$, the branch $\hat{B} = \max_b\{\hat{\alpha}_{d,b} > (c - \epsilon)\}$ and the number of nodes $N = \mathbb{V}_{node}\{\hat{\alpha}_{d,b} > (c - \epsilon)\}$. This leads to the optimal tree, governed by what we refer to as the **Greater Than Principle (GT principle)** in Equation equation 9, where $\hat{\alpha}_{d,b} > (c - \epsilon)$.

### 3.2 INTEGRATING GT PRINCIPLE INTO TREE-BASED ACCELERATION METHODS

We utilize the GT principle to construct tree structures for both static and dynamic frameworks. In static frameworks (Section 3.2.1), the tree dimensions are directly determined from pre-calculated probabilities. In dynamic frameworks (Section 3.2.2), the tree topology is regulated at runtime by evaluating node probabilities during expansion.

#### 3.2.1 STATIC TREE

In this section, we contrast our approach with Sequoia (Chen et al., 2024b). The fundamental difference lies in the optimization objective: Sequoia aims to maximize the average acceptance length, whereas our method explicitly seeks the optimal speedup ratio.

Sequoia employs dynamic programming to search for a tree structure that maximizes acceptance length within a broad, heuristic constraint space (e.g., node budget 128, max depth 10). However, maximizing length does not necessarily guarantee maximum speedup. Searching within such a wide range often results in inefficient nodes that increase computational overhead without providing proportional speed gains.

In contrast, our GT principle directly calculates the optimal tree geometry based on the cost coefficient $c$. First, the optimal depth $D$ is determined by the point where the cumulative acceptance probability of the primary path drops below the efficiency threshold:

$$D = \max_d \{\hat{\alpha}_{d,b} > (c - \epsilon)\} = \log_{\alpha_0} c + \epsilon \tag{10}$$

The optimal branching factor $B$ is determined by the count of viable tokens at each step:

$$B = \max_b \{\hat{\alpha}_{d,b} > (c - \epsilon)\} = Count\{[\hat{\alpha}_i > (c - \epsilon)]\} \tag{11}$$

Finally, the total number of nodes $S$ is strictly defined as the aggregate of all tokens that satisfy the GT condition, eliminating the need for a pre-defined node budget: $N = \sum_{d,b} \mathbb{I}(\hat{\alpha}_{d,b} > c - \epsilon)$. where $\mathbb{I}(\cdot)$ is the indicator function. By directly constructing the tree with these theoretically derived parameters ($D$, $B$, and $N$), we bypass the need for extensive searching and ensure the final structure delivers the maximum speedup ratio.

#### 3.2.2 DYNAMIC TREE

We extend the GT principle to EAGLE-2 (Li et al., 2024b) to address its structural inefficiencies through adaptive depth control and dynamic node allocation.

The standard EAGLE-2 relies on a rigid iterative process, typically involving 5 fixed iterations. In each step, it expands the top-10 nodes into 10 children each, followed by a global search to retain the top-60 candidates. This design imposes three critical constraints: (1) a fixed depth determined by iteration count, (2) a fixed branching factor, and (3) a static node budget. These limitations prevent the discovery of efficient "narrow-and-deep" structures and incur unnecessary computation on low-probability nodes.

Specifically, we determine the adaptive depth $D$ by monitoring the acceptance ratio of the most probable path (using the highest probability node at depth $i$, denoted as $\hat{\alpha}_{i,max}$, as a proxy):

$$D = \max_d \{\hat{\alpha}_{d,b} > (c - \epsilon)\} = \max_d \{\hat{\alpha}_{i,max} > (c - \epsilon)\} \tag{12}$$

To balance verification efficiency with coverage, we regulate the total node count $N$ via:

$$N = \min(D \times B, N_{max}) \tag{13}$$

where $B$ is the branching factor (kept consistent with the default EAGLE-2 setting) and $N_{max}$ is a hardware-imposed cap. This mechanism decouples EAGLE-2 from fixed depth and node count constraints. Unlike the original model, our approach adaptively halts expansion when acceptance probabilities drop, while extending generation deeper when confidence remains high. This achieves a simultaneous optimization of inference efficiency and structural flexibility.

Table 1: The result of efficiency. **Speed**: Inference efficiency improvement. $L$: Average acceptance length (quality metric). Configurations: **Vicuna-68M / Static Tree** and **EAGLE / Dynamic Tree** denote the draft model paired with static and dynamic verification trees, respectively. **GT(s) / GT(d)**: The static and dynamic versions of the GT principle. **ΔImp**: The performance improvement achieved by the proposed method.

| Task | | MT | | Trans | | Sum | | QA | | MR | | RAG | | Overall | |
|---|---|---|---|---|---|---|---|---|---|---|---|---|---|---|---|
| | | Speed | $L$ | Speed | $L$ | Speed | $L$ | Speed | $L$ | Speed | $L$ | Speed | $L$ | Speed | $L$ |
| **Vicuna-68M / Static Tree** | | | | | | | | | | | | | | | |
| Size | Tree | Speed | $L$ | Speed | $L$ | Speed | $L$ | Speed | $L$ | Speed | $L$ | Speed | $L$ | Speed | $L$ |
| 7B | Sequoia | 2.07× | 3.20 | 1.84× | 3.35 | 2.28× | 3.87 | 2.39× | 3.61 | 2.11× | 3.34 | 2.03× | 3.72 | 2.12× | 3.52 |
| | GT(s) | 2.17× | 2.96 | 1.92× | 3.12 | 2.32× | 3.53 | 2.41× | 3.29 | 2.18× | 3.31 | 2.11× | 3.41 | 2.19× | 3.27 |
| | ΔImp | **0.10×** | **-0.24** | **0.08×** | **-0.23** | **0.04×** | **-0.26** | **0.02×** | **0.32** | **0.07×** | **-0.03** | **0.08×** | **-0.31** | **0.07×** | **-0.25** |
| 13B | Sequoia | 1.81× | 3.07 | 1.60× | 2.57 | 2.14× | 3.68 | 1.96× | 3.18 | 2.09× | 3.30 | 1.94× | 3.54 | 1.92× | 3.23 |
| | GT(s) | 1.86× | 3.03 | 1.71× | 2.52 | 2.20× | 3.69 | 1.99× | 3.16 | 2.12× | 3.27 | 1.95× | 3.52 | 1.97× | 3.20 |
| | ΔImp | **0.05×** | **-0.04** | **0.11×** | **-0.05** | **0.06×** | **0.01** | **0.03×** | **-0.02** | **0.03×** | **-0.03** | **0.01×** | **-0.02** | **0.05×** | **-0.03** |
| 33B | Sequoia | 1.99× | 2.98 | 1.87× | 2.75 | 2.14× | 3.35 | 2.06× | 3.06 | 2.01× | 3.02 | 1.99× | 3.32 | 2.01× | 3.08 |
| | GT(s) | 2.01× | 2.97 | 1.84× | 2.70 | 2.24× | 3.34 | 2.16× | 3.05 | 2.01× | 3.02 | 2.05× | 3.33 | 2.05× | 3.07 |
| | ΔImp | **0.02×** | **-0.01** | **0.03×** | **-0.05** | **0.10×** | **-0.01** | **0.10×** | **-0.01** | **0.00×** | **0.00** | **0.06×** | **0.01** | **0.04×** | **-0.01** |
| **EAGLE / Dynamic Tree** | | | | | | | | | | | | | | | |
| Tree | Tree | Speed | $L$ | Speed | $L$ | Speed | $L$ | Speed | $L$ | Speed | $L$ | Speed | $L$ | Speed | $L$ |
| 7B | EAGLE-2 | 3.06× | 4.76 | 1.70× | 3.22 | 2.42× | 3.95 | 2.35× | 3.70 | 2.74× | 4.88 | 2.14× | 3.95 | 2.40× | 4.36 |
| | GT(d) | 3.17× | 5.09 | 2.01× | 3.17 | 2.57× | 3.95 | 2.46× | 3.74 | 2.81× | 5.04 | 2.23× | 3.96 | 2.54× | 4.52 |
| | ΔImp | **0.11×** | **0.33** | **0.31×** | **-0.05** | **0.15×** | **0** | **0.11×** | **0.04** | **0.07×** | **0.16** | **0.09×** | **0.01** | **0.14×** | **0.16** |
| 13B | EAGLE-2 | 2.85× | 4.80 | 2.06× | 3.32 | 2.54× | 4.13 | 2.26× | 3.52 | 3.32× | 4.80 | 2.30× | 4.25 | 2.56× | 4.43 |
| | GT(d) | 3.12× | 5.15 | 2.10× | 3.27 | 2.73× | 4.18 | 2.41× | 3.48 | 3.27× | 5.17 | 2.40× | 4.13 | 2.67× | 4.62 |
| | ΔImp | **0.27×** | **0.35** | **0.04×** | **-0.05** | **0.19×** | **0.05** | **0.15×** | **-0.04** | **-0.05×** | **0.37** | **0.10×** | **0.12** | **0.11×** | **0.19** |
| 33B | EAGLE-2 | 3.06× | 4.29 | 2.06× | 3.16 | 2.57× | 3.81 | 2.24× | 3.27 | 3.26× | 4.79 | 2.39× | 3.64 | 2.61× | 4.06 |
| | GT(d) | 3.11× | 4.44 | 2.19× | 3.13 | 2.63× | 3.81 | 2.33× | 3.25 | 3.37× | 5.06 | 2.40× | 3.60 | 2.68× | 4.15 |
| | ΔImp | **0.05×** | **0.15** | **0.13×** | **-0.03** | **0.06×** | **0.00** | **0.09×** | **-0.02** | **0.11×** | **0.27** | **0.01×** | **-0.04** | **0.07×** | **0.09** |

# 4 EXPERIMENT

## 4.1 EXPERIMENTAL SETTINGS

**Datasets and Models** We adopted the six subtasks from Spec-Bench (Xia et al., 2024), including multi-round conversations, translation, and more, which respectively utilize datasets like MT-bench (Zheng et al., 2023), WMT14 DE-EN, CNN/Daily Mail (Nallapati et al., 2016), Natural Questions (Kwiatkowski et al., 2019), GSM8K (Cobbe et al., 2021), and DPR (Karpukhin et al., 2020). For the base models, we chose Vicuna-v1.3 (Chiang et al., 2023) (with 7B, 13B, and 33B versions), LLama2 (Touvron et al., 2023b) (7B, 13B), Qwen2 (An Yang et al., 2024) (7B), and Llama3 (AAaron Grattafiori et al., 2024) (8B). To comprehensively assess our method, we compared it with two leading methods, the static tree Sequoia (Chen et al., 2024b) and the dynamic tree EAGLE-2 (Li et al., 2024b), both known for high speedup ratios. This setup enables us to clearly showcase how our method performs against state-of-the-art approaches across various model sizes. And the more method (Medusa (Cai et al., 2024), Hydra (Ankner et al., 2024), EAGLE (Li et al., 2024a), EAGLE-3 (Li et al., 2025), Tetris (Wu et al., 2025)) we conduct at Appendix A.3.

**Evaluation Metrics and Environment** We employ speedup ratios $Speed$ and average acceptance length $L$ as key metrics. The former evaluates the efficiency of the method by comparing the throughput (token/s) acceleration achieved by each method versus the vanilla approach, while the latter evaluates the quality of token generation by measuring the average number of tokens accepted. Most of the experiments were performed on a computing platform equipped with four NVIDIA GeForce RTX 4090 GPUs (24GB). To ensure comparability, we standardize hyperparameters, using greedy decoding, FP16, and BP16 precision. All experiments were repeated 5 times with the same hyperparameters and the results are reported as means.

## 4.2 THE STUDY OF EFFICIENCY

In this section, we examine how the GT principle impacts the efficiency of static and dynamic trees. Table 1 quantifies our method's efficiency gains across model scales. Our method delivers speedup gains of 0.07× (7B), 0.05× (13B), and 0.04× (33B), with corresponding $L$ reductions of 0.25, 0.03, and 0.04. Unlike the Sequoia method, which often yields excessively large trees to maximize accep-

tance length $L$, our GT principle optimizes tree size for computational efficiency. It achieves this by pruning low-return nodes during decoding, significantly accelerating inference. In dynamic tree configurations, we observe speedup increases of 0.14× (7B), 0.11× (13B), and 0.07× (33B), accompanied by $L$ enhancements of 0.16, 0.19, and 0.09. EAGLE-2's fixed maximum depth constrains tree growth, preventing the method from exploiting opportunities for longer valid sequences. Our GT solves this problem through an automatic depth adjustment mechanism and achieves consistent improvements in L across most tasks. This adaptive expansion excels in high-acceptance scenarios, enabling broader exploration and better performance than fixed-depth methods. In conclusion, our method consistently enhances efficiency across model sizes and decoding setups.

## 4.3 THE STUDY OF GENERALIZATION

To assess the generalization capability of our approach, we evaluated the performance of the GT principle on four different large language models, including the Llama-2 (7B, 13B), Llama-3 and qwen2. For fairness, we employed a standardized dataset (MT-bench), retained the same evaluation metrics of Experiment 4.2 (speed and L), and adopted the same parameter settings. The baseline method for comparison is the EAGLE-2. The results are listed in Table 2. GT achieves significant improvements, achieving speedups of 1.65-2.54×. In particular, on Llama-3-8B, GT achieved a

Table 2: Generalization Speedup Performance of Different LLMs on MT-Bench.

| Model | size | Origin | | GT | |
|-------|------|--------|-----|--------|-----|
| | | Speed | $L$ | Speed | $L$ |
| Llama-2 | 7B | 1.95× | 3.57 | **2.24×** | 3.60 |
| Llama-2 | 13B | 2.33× | 4.01 | **2.54×** | 4.03 |
| Qwen2 | 7B | 1.40× | 3.47 | **1.65×** | 4.03 |
| Llama-3 | 8B | 1.75× | 3.46 | **2.15×** | 3.41 |

0.40x improvement and a 2.15× speedup. This demonstrates its strong generalization capability and the effectiveness of optimizing inference across different model architectures.

## 4.4 THE STUDY OF MEMORY-TO-SPEEDUP RATIO

Recent advances in speculative sampling are gaining attention for speeding up LLM inference. However, these methods often increase memory usage, which limits their efficiency in resource-constrained settings like edge cloud computing and IoT devices (Svirschevski et al., 2024; Xu et al., 2024). In this study, we measured speedup ratios and GPU memory usage (GB) for both baseline and our method. To quantify the speedup-memory trade-off, we use $R = \frac{Speed}{GPU(GB)}$, where a higher $R$ indicates more efficient resource use under fixed memory. Table 3 shows our method has near-identical GPU memory usage to the baseline while achieving substantial speedup gains.

Table 3: Memory Utilization ($R$).

| Size | Model | Speed | GPU(GB) | $R$ |
|------|-------|-------|---------|-----|
| 7B | Vanilla | 1.00× | 14.66773 | 0.06817 |
| | EAGLE-2 | 2.40× | 16.42108 | 0.14615 |
| | GT | 2.54× | 16.42126 | **0.15467** |
| 13B | Vanilla | 1.00× | 27.56194 | 0.03628 |
| | EAGLE-2 | 2.56× | 29.61629 | 0.08643 |
| | GT | 2.67× | 29.61633 | **0.09015** |
| 33B | Vanilla | 1.00× | 67.46208 | 0.01482 |
| | EAGLE-2 | 2.61× | 72.33713 | 0.03608 |
| | GT | 2.68× | 72.33717 | **0.03704** |

Consequently, our method yields a higher $R$, demonstrating superior resource efficiency. These results confirm that it delivers speedup benefits with minimal extra memory overhead, highlighting advantages in both resource efficiency and performance.

## 4.5 THE STUDY OF SENSITIVITY

### 4.5.1 IMPACT OF DIFFERENT THRESHOLDS

Although the error $\epsilon = |\alpha^* - \hat{\alpha}|$ between true and estimated acceptance probabilities is unobservable during inference, we approximate it by quantifying the discrepancy between the empirically optimal threshold $\hat{c}$ and the theoretically derived threshold $c$. To validate this relationship, we assessed thresholds ranging from 0.01 to 0.40 on MT-bench with EAGLE-2.

Table 4: Performance at different $c$ on Vicuna-7b-v.13 (token/s and length).

| $c$ | 0.01 | 0.05 | 0.10 | 0.109 | 0.110 | 0.111 | 0.12 | 0.15 | 0.20 | 0.30 | 0.40 |
|---|---|---|---|---|---|---|---|---|---|---|---|
| Speed | 77.60 | 89.74 | 91.92 | 93.92 | **95.55** | 94.82 | 93.55 | 91.92 | 88.30 | 86.97 | 83.79 |
| $L$ | 5.57 | 5.41 | 5.29 | 5.28 | 5.25 | 5.23 | 5.21 | 5.13 | 5.06 | 4.79 | 4.40 |

As shown in Table 4, optimal throughput performance (95.55 token/s) was achieved at a threshold $\hat{c}$ of 0.110. Our theoretical calculation of $c = \frac{M_D}{M_T}$ yields 0.109, resulting in an estimated $\epsilon$ of only 0.001, which is an extremely small deviation. This indicates that the closer the estimated acceptance probability is to the true acceptance probability, the more accurate our method becomes. Overall, this further confirms the rationale of the proposed GT principles, which are very valuable guidelines with practical applicability.

### 4.5.2 IMPACT OF DIFFERENT BATCH SIZES

Most existing speculative sampling methods are tailored for a batch size of 1. Unfortunately, EAGLE-2 and Sequoia's official implementations lack native support for large batches, and modifying their vanilla code for multi-batch deployment poses challenges. To address this, we explored various alternative approaches and ultimately succeeded in integrating the GT principle (coupled with EAGLE-2) into SGLang (Zheng et al., 2024), which facilitates flexible batch processing. With

Table 5: Throughput performance (tokens/s) for different batch sizes on LLaMA3.1-70B.

| Batch Size | 2 | 4 | 8 | 16 | 32 | 64 |
|---|---|---|---|---|---|---|
| EAGLE-2 | 43.38 | 77.42 | 124.51 | 188.46 | 194.82 | 197.96 |
| GT | 44.24 | 80.92 | 146.95 | 197.95 | 203.71 | 208.35 |

this setup, we evaluated LLaMA3.1-70B across batch sizes on 4 A6000 GPUs (48GB); throughput (tokens/s) is in Table 5. For batch sizes $\leq 16$, speculative sampling achieves reasonable speedup, but for batch sizes $> 32$, the speedup decreases due to key-value cache overflow. Nevertheless, GT, with its dynamic adaptability, consistently outperforms EAGLE-2. This effectiveness is supported by the results of Experiment 4.4: it boosts speedup without extra memory.

### 4.5.3 IMPACT OF DIFFERENT SAMPLE TEMPERATURES

To assess the stability of the proposed method, we examined the model's acceleration performance across different temperature configurations. Because temperature randomness during token sampling can affect the model's acceleration, the acceleration effect of speculative sampling weakens with increasing temperature. Therefore, we compared our method with EAGLE-2 under the same temperature conditions. Results in Figure 3 demonstrate that GT outperforms EAGLE-2 on both Llama-2-7B and Vicuna-7B architectures as temperature increases, sustaining a 0.2×-0.3× speedup

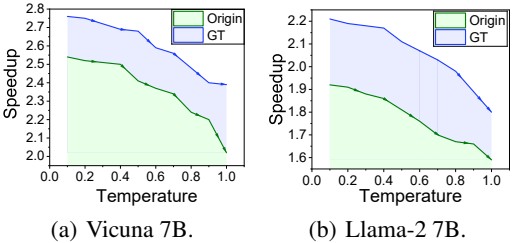

(a) Vicuna 7B.  (b) Llama-2 7B.

Figure 3: GT at different sample temperatures.

performance gain. In summary, these findings confirm the robustness of the GT principle, validate its efficacy in enhancing both stability and performance, and highlight its potential for deployment in scenarios susceptible to randomness.

### 4.6 CASE STUDY

To clarify the acceptance of the draft and highlight the advancement of our approach over previous work, we conduct a case study. As shown in Figure 4, GT accepts longer, more contiguous token sequences, while Origin (EAGLE-2), constrained by fixed draft length, terminates prematurely. In the first underlined segment: Origin needs two speculative sampling rounds. "Students will be able to" is accepted firstly, and then the second "the" in a subsequent round. By contrast, GT completes sampling in one round, generating the contiguous accepted sequence ("students will be able to understand the") cohesively. A similar pattern appears in the second underlined segment: "and" is integrated into GT's draft in one drafting step. These observations confirm that our method uses adaptive draft length, enabling continuous drafting until generation meets termination criteria. More tree cases and output results are in Appendix A.7.

Figure 4: Case Study. The highlighted tokens are from the draft model and the underlined words show the differences between the two methods.

## 5 RELATED WORK

Speculative sampling achieves lossless acceleration via original LLM verification (Zhang et al., 2023; Stern et al., 2018; Chen et al., 2023). SpecDec (Xia et al., 2023) pioneered independent draft models, and (Leviathan et al., 2023) used T5-small to speed up T5-XXL. These lightweight pre-trained models need no extra training (Spector & Re, 2023; Sun et al., 2024), but their adaptability across architectures needs improvement. Recent work optimizes tree structures for efficiency: SpecInfer (Miao et al., 2024) introduced token tree verification; Sequoia (Chen et al., 2024b) has hardware-adaptive trees; OPT-Tree (Wang et al., 2025) uses search for optimal topologies; DSBD (Qin et al., 2024) has a two-stage pipeline; DySpec (Xiong et al., 2025) uses confidence-guided expansion; EAGLE-2 (Li et al., 2024b) adds context-sensitivity. DD (Brown et al., 2024) improved the draft tree generation, and PCT (Zheng & Wang, 2025)) pruned away redundant nodes. In hybrid methods combining tree-based verification with other techniques: ProPD (Zhong et al., 2024) integrated progressive refinement into hierarchical structures, and RSD (Jeon et al., 2024) proposed recursive verification mechanisms; GSD (Gong et al., 2024) and ADED (Liu et al., 2025) extended traditional tree models via graph-based representations and adaptive depth control to handle complex dependencies. For parallel multi-draft verification, (Hu et al., 2025) proposed a hybrid sampling strategy to improve acceptance in specific scenarios, while (Khisti et al., 2024) introduced a two-phase framework to optimize parallel draft generation. And more detail is in Appendix A.2. However, existing approaches prioritize maximizing the average acceptance length, wasting resources on low-reward nodes and missing optimal trees. To address this gap, we propose a GT principle for tree construction, where the primary objective is to achieve the highest acceleration efficiency.

## 6 CONCLUSION AND FUTURE WORK

In this paper, we reexamine the relationship between draft length and expected speedup, discovering the difficulty of analytical analysis and strong the node homogeneity assumption assumptions. We then redefine the cost and benefit of each node, establish a functional relationship between time reduction draft length, and prove convexity. Furthermore, we extend this principle to trees and propose a universal tree construction principle applicable to various labeled tree verification methods, aiming to maximize speedup. Experiments on benchmark datasets deliver significant gains, improving overall performance of 4% to 14% and achieving end-to-end speedup of 1.97× to 2.68×. Extensive experiments further confirm the effectiveness, stability, and versatility of our method. These findings redefine the goal of tree construction in speculative sampling and open up a promising direction for future research. However, this work centers on algorithmic optimizations; practical implementations typically require integration with system-level acceleration techniques. Going forward, this research will explore the integration of additional inference frameworks (such as vLLM and SGLang) to unify algorithmic, system-level, and hardware-level optimizations.

# 7 ETHICS STATEMENT

All authors of this paper have carefully read and strictly adhere to the ICLR Code of Ethics. We confirm that all aspects of our research, including paper submission, data collection, experimental design, and result reporting, fully comply with the ethical requirements specified in the code.

# 8 REPRODUCIBILITY STATEMENT

To ensure the reproducibility of our research results, we have made the following targeted efforts: First, we have uploaded the anonymous source code of the proposed model. Second, detailed information about the experimental setup is clearly described in Section 4.1. Third, for the theoretical results presented in this paper, complete proof processes and derivation details are included in Section 3.1 and Appendix A.9, which can be referenced to verify the validity of the theoretical claims. All the above materials are mutually complementary, providing a comprehensive basis for the reproducibility of our research.

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

# A  APPENDIX

## A.1  THE USAGE OF LLMS

In this paper, we only use LLMs to help us polish the text, including correcting grammar and checking word usage.

## A.2  RELATED WORK

### A.2.1  SPECULATIVE SAMPLING

As large language models have become widely adopted, numerous methods have been proposed to speed up LLM inference, such as low-bit quantization (Hubara et al., 2018; Shen et al., 2020; Kim et al., 2021), pruning (Gale et al., 2019), and knowledge distillation (Hinton, 2015). These techniques reduce generation latency by cutting down the computational cost of each forward pass. However, they often lead to a trade-off, as LLM performance can suffer in exchange for improved computational efficiency.

Speculative sampling methods achieve lossless acceleration by using the original LLM for verification. (Xia et al., 2022; Zhang et al., 2023) The draft-then-verify decoding strategy, introduced by Stern et al. (Stern et al., 2018), laid the foundation for speculative sampling (Chen et al., 2023), which adapts this approach to non-greedy sampling while preserving the original output distribution. SpecDec (Xia et al., 2023) pioneered the use of independent models for drafting, striking a balance between accuracy and efficiency. Subsequent work by Leviathan et al.(Leviathan et al., 2023) accelerated T5-XXL inference by using a smaller T5-small model for drafting. These methods leverage lightweight, pre-trained models that do not require additional training, enabling seamless adoption in various applications (Spector & Re, 2023; Sun et al., 2024). MagicDec (Chen et al., 2024a) explore different KV compression algorithms for drafting and present a bottleneck-aware general formulation to select suitable drafting strategy based on task, batch-size and sequence-length. QuantSpec (Tiwari et al., 2025) propose a double full-precision cache buffer to resolve conflicts between per-group quantization and speculative sampling. Judge Decoding (Bachmann et al., 2025) proposed an adapted verification scheme that makes use of the capability of LLMs to judge responses in a versatile way.

Inference sampling improves efficiency, but a careful balance needs to be struck between speed and accuracy of the draft model to ensure optimal results. However, further improvements are needed to increase its adaptability and robustness across different model architectures.

### A.2.2  TOKEN TREE VERIFICATION

Recent studies have centered on optimizing tree structures and sizes to enhance computational efficiency. SpecInfer (Miao et al., 2024) introduced token tree verification, refining earlier strategies that only verified single draft sequences. Sequoia (Chen et al., 2024b) developed a hardware-adaptive tree optimization framework, dynamically selecting tree dimensions according to available computing resources to maximize inference speed. OPT-Tree (Wang et al., 2025) adopted a search-based approach for optimal tree topologies, aiming to maximize the expected accepted token length per decoding step. DSBD (Qin et al., 2024) employed a two-stage pipeline: a lightweight model generated beam search candidates, which were then verified layer-by-layer by a large model with dynamic beam width adjustment based on acceptance probabilities, balancing efficiency and quality. DySpec (Xiong et al., 2025) enabled real-time tree expansion guided by prediction confidence, while EAGLE-2 (Li et al., 2024b) incorporated context-sensitive tree construction to boost acceptance rates. Building on EAGLE-2, DDD (Brown et al., 2024) improved the draft tree generation by making depth selection contingent on model confidence scores, and PCT (Zheng & Wang, 2025) pruned away redundant nodes.

In the realm of hybrid methodologies combining tree-based verification with complementary techniques, ProPD (Zhong et al., 2024) integrated progressive refinement into hierarchical structures, and RSD (Jeon et al., 2024) proposed recursive verification mechanisms. GSD (Gong et al., 2024) and ADED (Liu et al., 2025) extended traditional tree models using graph-based representations and adaptive depth control to handle complex dependency structures. For parallel multi-draft verification

Table 6: The result of efficiency.

| Task | | MT | | Trans | | Sum | | QA | | MR | | RAG | | Overall | |
|---|---|---|---|---|---|---|---|---|---|---|---|---|---|---|---|
| | | Speed | $L$ | Speed | $L$ | Speed | $L$ | Speed | $L$ | Speed | $L$ | Speed | $L$ | Speed | $L$ |
| **EAGLE** | | | | | | | | | | | | | | | |
| Size | Model | Speed | $L$ | Speed | $L$ | Speed | $L$ | Speed | $L$ | Speed | $L$ | Speed | $L$ | Speed | $L$ |
| 7B | Base | 2.63× | 3.84 | 1.81× | 2.83 | 2.24× | 3.31 | 2.19× | 3.13 | 2.33× | 3.81 | 1.90× | 3.20 | 2.18× | 3.55 |
| | GT | 2.97× | 4.42 | 1.87× | 2.96 | 2.33× | 3.58 | 2.24× | 3.38 | 2.67× | 4.44 | 2.06× | 3.65 | 2.36× | 4.01 |
| | ΔImp↑ | **0.34×** | **0.58** | **0.06×** | **0.16** | **0.09×** | **0.27** | **0.05×** | **0.25** | **0.34×** | **0.63** | **0.16×** | **0.45** | **0.18×** | **0.46** |
| 13B | Base | 2.59× | 3.89 | 2.03× | 2.94 | 2.43× | 3.44 | 1.96× | 2.94 | 2.72× | 3.93 | 2.10× | 3.54 | 2.31× | 3.64 |
| | GT | 2.91× | 4.47 | 2.13× | 3.07 | 2.40× | 3.76 | 2.01× | 3.12 | 3.05× | 4.54 | 2.44× | 3.95 | 2.50× | 4.09 |
| | ΔImp↑ | **0.32×** | **0.58** | **0.10×** | **0.13** | **0.03×** | **0.32** | **0.05×** | **0.18** | **0.33×** | **0.61** | **0.34×** | **0.41** | **0.19×** | **0.45** |
| 33B | Base | 2.64× | 3.52 | 1.90× | 2.80 | 2.38× | 3.25 | 2.08× | 2.82 | 3.01× | 3.88 | 1.89× | 3.05 | 2.32× | 3.38 |
| | GT | 2.87× | 3.93 | 1.98× | 2.93 | 2.37× | 3.52 | 2.08× | 2.96 | 3.10× | 4.45 | 2.15× | 3.29 | 2.43× | 3.72 |
| | ΔImp↑ | **0.23×** | **0.41** | **0.08×** | **0.13** | **-0.01×** | **0.27** | **0.00×** | **0.14** | **0.09×** | **0.57** | **0.26×** | **0.24** | **0.11×** | **0.34** |
| **MEDUSA** | | | | | | | | | | | | | | | |
| Size | Model | Speed | $L$ | Speed | $L$ | Speed | $L$ | Speed | $L$ | Speed | $L$ | Speed | $L$ | Speed | $L$ |
| 7B | Base | 1.86× | 2.51 | 1.53× | 2.12 | 1.55× | 2.01 | 1.49× | 2.04 | 1.71× | 2.52 | 1.28× | 2.08 | 1.57× | 2.31 |
| | GT | 2.04× | 2.55 | 1.71× | 2.13 | 1.65× | 2.02 | 1.62× | 2.06 | 1.57× | 2.51 | 1.32× | 2.11 | 1.64× | 2.33 |
| | ΔImp↑ | **0.18×** | **0.04** | **0.18×** | **0.01** | **0.10×** | **0.01** | **0.13×** | **0.02** | **-0.14×** | **-0.01** | **0.04×** | **0.03** | **0.07×** | **0.02** |
| 13B | Base | 1.42× | 2.58 | 1.23× | 2.19 | 1.02× | 2.09 | 1.21× | 2.10 | 1.46× | 2.59 | 1.01× | 2.12 | 1.23× | 2.39 |
| | GT | 1.50× | 2.61 | 1.29× | 2.21 | 1.10× | 2.10 | 1.27× | 2.12 | 1.55× | 2.62 | 1.08× | 2.13 | 1.30× | 2.41 |
| | ΔImp↑ | **0.08×** | **0.03** | **0.06×** | **0.02** | **0.08×** | **0.01** | **0.06×** | **0.02** | **0.09×** | **0.03** | **0.07×** | **0.01** | **0.07×** | **0.02** |
| **HYDRA** | | | | | | | | | | | | | | | |
| Size | Model | Speed | $L$ | Speed | $L$ | Speed | $L$ | Speed | $L$ | Speed | $L$ | Speed | $L$ | Speed | $L$ |
| 7B | Base | 2.45× | 3.59 | 1.97× | 2.79 | 1.92× | 2.70 | 1.91× | 2.91 | 2.30× | 3.66 | 1.64× | 2.90 | 2.04× | 3.26 |
| | GT | 2.57× | 3.69 | 2.04× | 2.84 | 1.96× | 2.74 | 1.99× | 2.95 | 2.41× | 3.75 | 1.68× | 2.93 | 2.12× | 3.33 |
| | ΔImp↑ | **0.12×** | **0.10** | **0.07×** | **0.05** | **0.04×** | **0.04** | **0.08×** | **0.04** | **0.09×** | **0.09** | **0.04×** | **0.03** | **0.08×** | **0.07** |
| 13B | Base | 1.82× | 3.65 | 1.45× | 2.81 | 1.26× | 2.82 | 1.50× | 2.88 | 1.89× | 3.70 | 1.28× | 3.08 | 1.54× | 3.35 |
| | GT | 1.90× | 3.72 | 1.50× | 2.86 | 1.33× | 2.86 | 1.56× | 2.91 | 1.95× | 3.79 | 1.33× | 3.23 | 1.60× | 3.42 |
| | ΔImp↑ | **0.08×** | **0.07** | **0.05×** | **0.05** | **0.07×** | **0.04** | **0.06×** | **0.03** | **0.06×** | **0.09** | **0.05×** | **0.15** | **0.06×** | **0.07** |
| **EAGLE-3** | | | | | | | | | | | | | | | |
| Size | Model | Speed | $L$ | Speed | $L$ | Speed | $L$ | Speed | $L$ | Speed | $L$ | Speed | $L$ | Speed | $L$ |
| 13B | Base | 3.49× | 5.98 | 2.38× | 4.30 | 3.26× | 5.76 | 2.91× | 4.82 | 3.40× | 5.78 | 2.80× | 5.69 | 3.04× | 5.69 |
| | GT | 3.76× | 8.38 | 2.42× | 4.40 | 3.45× | 7.88 | 2.94× | 5.31 | 3.44× | 7.75 | 2.88× | 7.11 | 3.15× | 7.48 |
| | ΔImp↑ | **0.27×** | **2.40** | **0.04×** | **0.10** | **0.19×** | **2.12** | **0.03×** | **0.49** | **0.04×** | **1.97** | **0.08×** | **1.42** | **0.11×** | **1.79** |

(MDSD), (Hu et al., 2025) proposed a hybrid sampling strategy that deterministically selected high-probability tokens while stochastically sampling final candidates, improving acceptance in specific scenarios. (Khisti et al., 2024) introduced a two-phase framework: importance sampling was first used to select draft tokens, followed by single-draft verification, optimizing parallel draft generation workflows.

However, existing approaches universally prioritize maximizing the average acceptance length, a strategy that incurs excessive temporal and resource costs when exploring low-reward nodes, fundamentally impeding the discovery of globally optimal verification trees. Consequently, a critical void persists in establishing a general principle for constructing trees with optimal speedup ratios. In this paper, we introduce the Generic Tree Construction Principle, a systematic framework for optimizing tree structures to achieve minimal inference time.

## A.3 ADDITION EXPERIMENT

## A.4 EFFICIENCY OF ADVANCED METHODS

We tried other tree methods (EAGLE, EAGLE-3, Mudesa and Hydra), and due to the length limitation of the main paper, we present the results at table 6. For EAGLE, we achieve additional speedups ranging from 0.11× to 0.18× with additional average acceptance length improvement from 0.34 to 0.46 on models ranging from 7B to 33B. Meanwhile, GT continued to accelerate on medusa and hydra, achieving additional speedups of 0.7× and form 0.6 to 0.8×, respectively, and the acceptance length increases of 0.2 and 0.7. From the experimental results, we can see that the token trees of current advanced methods have not reached their limits, and the GT principle can help them further expand on static trees. Notably, GT achieved a significant increase in acceptance length (1.79) with EAGLE-3 on 13B models, further improving the speedup (0.11×). This result shows that although

Table 7: Throughput performance (tokens/s) for Comparison with Medusa+ (Dynamic Tree Variant)

| Method | Time (seconds) | tokens | Speed (token/s) | L |
|---|---|---|---|---|
| Medusa+ | 21.125 | 1381 | 65.37 | 2.637 |
| GT | 18.491 | 1288 | 69.66 | 2.456 |
| ΔImp | -2.634 | -93 | 4.61 | -0.181 |

EAGLE-3 further improves the ability of the draft model in some aspects, its customized maximum length (6) is not suitable for all tasks. For example, on the MT, Sum, MR, RAG tasks, EAGLE-3, limited by its maximum length, fails to fully exploit the outstanding capabilities of the drafting model, whereas GT can. Furthermore, we conducted a specific comparison with the dynamic tree variant of MEDUSA (referred to as "Medusa+" (Zhang, 2025)). Since "Medusa+" fixes the drafting head and the maximum number of nodes, we applied our GT principle by simply selecting nodes with probabilities greater than our cost ratio (set to 0.01 in this case) from its original top-64 nodes. This approach effectively prunes less beneficial nodes, thereby reducing exploration and verification time. Comparison results are shown in Table 7. As shown, our GT principle consistently improves performance in both static and dynamic MEDUSA variants, achieving a 7% speedup improvement, further demonstrating its effectiveness in adaptive tree construction.

## A.5 MEMORY-TO-SPEED RATIO OF ADVANCED METHODS

To further explore the advantages of the GT principle in memory efficiency, we also explored its memory-to-speedup effect on the static tree EAGLE and the dynamic tree EAGLE-2, as shown in the table 8. The results show that, for both static and dynamic trees, the GT principle can achieve a significant speedup compared to the original method without requiring much additional memory, further demonstrating the effectiveness of our method.

Table 8: Memory Utilization ($R$).

| Size | Model | Speed | GPU(GB) | $R$ |
|---|---|---|---|---|
| 7B | Vanilla | 1.00× | 14.66773 | 0.06817 |
| | EAGLE | 2.18× | 16.43378 | 0.13265 |
| | EAGLE+GT | 2.36× | 16.43380 | **0.14360** |
| | EAGLE-2 | 2.40× | 16.42108 | 0.14615 |
| | EAGLE-2+GT | 2.54× | 16.42126 | **0.15467** |
| 13B | Vanilla | 1.00× | 27.56194 | 0.03628 |
| | EAGLE | 2.31× | 29.63288 | 0.08381 |
| | EAGLE+GT | 2.50× | 29.63290 | **0.08436** |
| | EAGLE-2 | 2.56× | 29.61629 | 0.08643 |
| | EAGLE-2+GT | 2.67× | 29.61633 | **0.09015** |
| 33B | Vanilla | 1.00× | 67.46208 | 0.01482 |
| | EAGLE | 2.32× | 72.35852 | 0.03206 |
| | EAGLE+GT | 2.43× | 72.35854 | **0.03358** |
| | EAGLE-2 | 2.61× | 72.33713 | 0.03608 |
| | EAGLE-2+GT | 2.68× | 72.33717 | **0.03704** |

## A.6 LARGE BATCH ON SEQUENCE-BASED METHOD

more significant. This is because the draft tree needs more computation and memory resources in such scenarios. Fortunately, the principle we propose is not only applicable to draft trees but also to multi-batch sequence scenarios. We conducted preliminary experiments on Tetris (Wu et al., 2025). Experimental setup for this comparison is as follows: we used 4 NVIDIA GeForce RTX 4090 GPUs, with the base model being vicuna-7b-v1.3 and the speculative model being vicuna-68m. For Tetris,

we set the batch size to 2, repeated the runs 3 times, and set the number of speculative tokens to 7. Under this setting, the throughput improvement and total time reduction are as follows:

Table 9: Throughput performance (tokens/s) for Comparison with Tetris.

| Method | Time (seconds) | Throught |
|---|---|---|
| Baseline | 88.27 | 1161.45 |
| Tetris | 83.03 | 1235.33 |
| **ΔTetris** | **1.0631** | **1.0636** |
| GT | 79.91 | 1276.13 |
| **ΔGT** | **1.1046** | **1.0987** |

As shown in the table 9, our approach improved throughput by 9.8% compared to the baseline, outperforming the original TETRIS's 6.3% improvement.

### A.7 TREE CASE AND OUTPUT RESULTS

The [Tree case]illustrates the difference between EAGLE-2 and GT. When the final accepted length is 4, EAGLE-2 maintains a fixed depth of 6 and contains 60 nodes at Figure 5. In contrast, due to a lower estimated acceptance probability, GT prunes the tree and stops at the fourth layer, resulting in only 40 nodes retained at Figure 6.

The [Output report] highlights the differences in output between EAGLE-2 and GT. In a generation task, GT requires fewer decoding steps—[32, 52] compared to EAGLE-2's [40, 56]—leading to a shorter total runtime ([1.967, 2.764] < [1.789, 2.62]). Moreover, the accepted draft lengths per decoding step show that GT can accept drafts longer than 7 tokens, reaching up to 20 in some cases, whereas EAGLE-2's maximum accepted draft length is limited to 7.

#### A.7.1 TREE CASE

[Tree case]: '0 1 2' ... is the number of each node; '0' is the root node, '1' and '2' are the child nodes of '0'; the vertical direction is the width, and the horizontal direction is the depth.

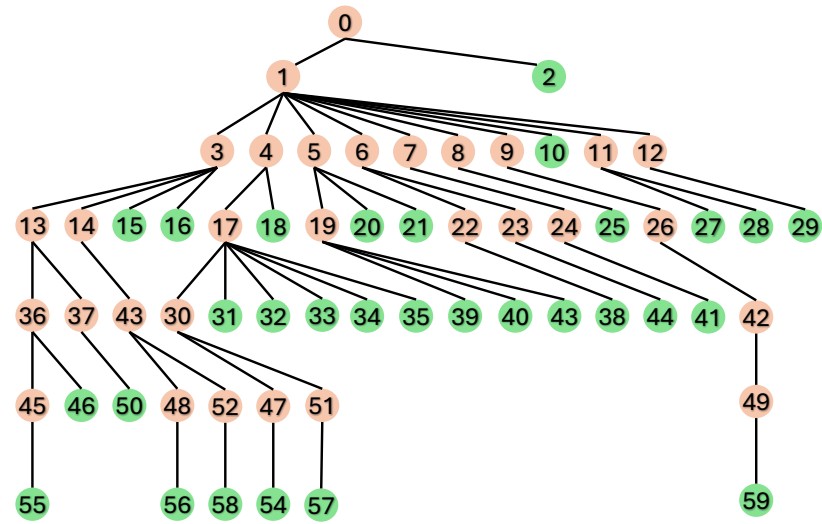

Figure 5: EAGLE-2 tree case.

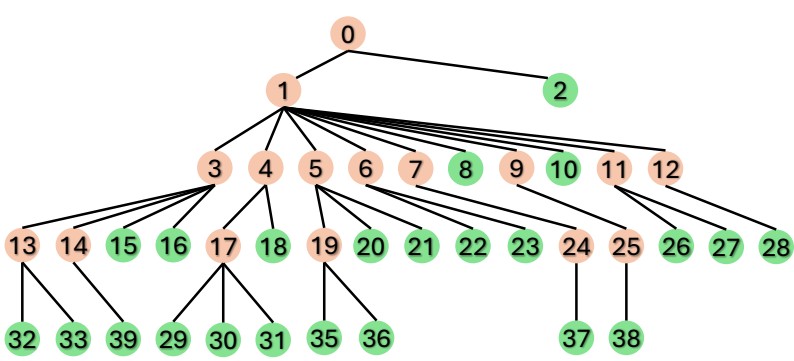

Figure 6: GT tree case.

### A.7.2 OUTPUTS RESULTS

[Output report]:'accept lengths' refers to the number of tokens generated in each decoding process. 'decoding steps' denotes the number of decoding operations between two generation processes. 'wall time' represents the time elapsed between two generation processes.

"Input":["Subject: Request for Feedback on Quarterly Financial Report. Dear [Supervisor's Name], I hope this email finds you well...","As an AI language model, I do not have personal opinions or emotions, so I cannot evaluate or critique my own response..."]

EAGLE-2: 'accept lengths': [3, 6, 7, 7, 7, 7, 5, 4, 7, 6, 1, 6, 6, 7, 2, 5, 7, 7, 5, 4, 5, 6, 7, 6, 3, 6, 7, 7, 7, 3, 3, 4, 4, 7, 4, 5, 7, 7, 7, 2, 4, 4, 3, 6, 4, 7, 3, 1, 3, 7, 2, 2, 3, 7, 5, 5, 7, 3, 4, 7, 4, 2, 2, 2, 4, 6, 7, 5, 7, 7, 5, 4, 5, 5, 7, 7, 5, 7, 2, 2, 6, 2, 5, 4, 3, 7, 7, 6, 4, 5, 6, 7, 5, 3, 2, 6], 'decoding steps': [40, 56], 'wall time': [1.967, 2.764] .

GT: 'accept lengths': [4, 6, 13, 14, 5, 4, 9, 4, 1, 6, 6, 9, 5, 7, 10, 6, 5, 6, 9, 7, 6, 7, 7, 10, 3, 4, 4, 7, 3, 6, 20, 3, 4, 4, 3, 4, 4, 3, 7, 3, 3, 7, 2, 2, 3, 8, 3, 5, 11, 4, 7, 4, 2, 2, 2, 4, 5, 6, 7, 7, 7, 5, 4, 5, 5, 18, 8, 8, 4, 5, 4, 3, 9, 10, 4, 6, 6, 6, 2, 4, 3, 2, 4, 2], 'decoding steps': [32, 52], 'wall time': [1.789, 2.62] .

### A.8 IMPLEMENT DETAILS

The parameter $c$ represents the ratio of wall-clock time between the draft model and the target model ($M_D/M_T$). For independent draft model and target model (e.g., Sequoia), $c$ is directly measured as the ratio of their generation times for a fixed number of tokens. When the draft model is a subcomponent of the target model (e.g., EAGLE-2), it is difficult to measure the standalone wall-clock time of the draft model. Therefore, we attempt to calculate the coefficient of $c$ by calculating the ratio of their parameter sizes. The specific $c$ values used for our experiments are detailed at 10:

Table 10:

| Method | Model | Draft | c |
|--------|-------|-------|---|
| EAGLE-2 | vicuna-7b-v1.3 | EAGLE-Vicuna-7B-v1.3 | 0.11 |
| EAGLE-2 | vicuna-13b-v1.3 | EAGLE-Vicuna-13B-v1.3 | 0.08 |
| EAGLE-2 | vicuna-33b-v1.3 | EAGLE-Vicuna-33B-v1.3 | 0.05 |
| Sequoia | vicuna-7b-v1.3 | vicuna-68m | 0.01 |
| Sequoia | vicuna-13b-v1.3 | vicuna-68m | 0.005 |
| Sequoia | vicuna-33b-v1.3 | vicuna-68m | 0.002 |

### A.9 Proof of Principle

This part mainly aims to prove the correctness of our method. First, let's review the basic concepts. In (Leviathan et al., 2023), the expected improvement factor (or expected speedup) in total wall-clock time:

$$\mathbb{E}(\text{Spd.}) = \frac{1 - \alpha^{\gamma+1}}{(1 - \alpha)(\gamma c + 1)}, \alpha \in (0, 1), c \in (0, 1). \tag{14}$$

Here, $\alpha$ is the average acceptance rate. The proposal length $\gamma$ denotes the drafting length of speculative sampling, and the cost coefficient $c$ is the ratio between the wall-clock time of a single run of the draft model $M_D$ and that of the target model $M_T$.

We define the cost of drafting the $i$ token as:

$$M_D + \Phi_{M_T}(\gamma), \quad \text{if } \gamma < K, \text{ then } \Phi_{M_T}(\gamma) = 0. \tag{15}$$

$\Phi_{M_T}(\gamma)$ is a function related the target model to verifying $\gamma$ tokens, which is usually equal to 0 up to a certain threshold $K$. And if a drafted token is accept, the system saves $M_T$ wall-clock time. Thus, we define the benefit as:

$$M_T \cdot (\hat{\alpha}_i + \epsilon) \tag{16}$$

$\hat{\alpha}_i$ denotes the estimate of the acceptance probability for the $i$-th token, while $\epsilon$ represents the error between the true acceptance probability ($\alpha^*$) and its estimate ($\hat{\alpha}$). The average acceptance rate ($\alpha$) used in formula 1 is the average probability of each node being accepted in its current state, while the acceptance rate ($\alpha_i$) we use is the probability of each node being accepted in the end. Thus, when we use the average acceptance rate $\alpha$ from formula 1 instead of analyzing them individually ($\hat{\alpha}_i = \prod \alpha_i = (\alpha)^i$), we can obtain the benefit as at one speculative sampling step:

$$\sum_0^\gamma \{M_T \cdot (\hat{\alpha}_i + \epsilon)\} = M_T \sum_0^\gamma (\alpha)^i + \epsilon = M_T \frac{1 - \alpha^{\gamma+1}}{1 - \alpha} + \epsilon \tag{17}$$

And, the time cost contains the base time $M_T$ and addition drafting time $M_D \cdot \gamma + \Phi_{M_T}(\gamma)$:

$$M_T + M_D \cdot \gamma + \Phi_{M_T}(\gamma) \tag{18}$$

If $\gamma < K$, then $\Phi_{M_T}(\gamma) = 0$. So, we can get the expected speedup as:

$$\mathbb{E}(\text{Spd.}) = \frac{M_T \frac{1 - \alpha^{\gamma+1}}{1 - \alpha} + \epsilon}{M_T + M_D \cdot \gamma} = \frac{\frac{1 - \alpha^{\gamma+1}}{1 - \alpha}}{1 + \frac{M_D}{M_T} \cdot \gamma} + \epsilon = \frac{1 - \alpha^{\gamma+1}}{(1 - \alpha)(\gamma c + 1)} + \epsilon \tag{19}$$

So, it shows that the cost and benefit is the single part of each drafting step. And the finally expected speedup formula is same as the 1.

