# OpenReview forum: "Max-Speedup Speculative Sampling: A Generic Tree Construction Principle"
_ICLR.cc/2026/Conference — ICLR 2026 Conference Withdrawn Submission_

### Official Review · Reviewer_PxyG · 2025-10-22

**Soundness:** 1
**Presentation:** 1
**Contribution:** 1
**Rating:** 2
**Confidence:** 4

**Summary:**

This paper investigates speculative sampling for accelerating large language model inference.
The authors propose a Generic Tree Construction Principle to decide which candidate nodes should be added during tree-based speculative decoding. They provide a convex-optimization-style analysis of expected time vs. acceptance rate and claim that the derived principle can yield maximum speedups under a given cost budget.

**Strengths:**

The attempt to formalize benefit-cost trade-offs in tree-based speculative decoding is interesting.

**Weaknesses:**

1 The proposed “generic principle” essentially reformulates existing tree-based speculative sampling as an optimization problem with pruning rules. It does not introduce a fundamentally new mechanism or algorithmic insight beyond prior work (e.g., Sequoia, EAGLE-2, SpecInfer). The contribution is more of an incremental improvement rather than a conceptual breakthrough.
2 The analysis relies on assuming that each node’s acceptance probability can be estimated accurately and independently. In real LLM decoding, acceptance rates depend on context, sampling temperature, and model randomness—these assumptions rarely hold.
The convexity proof therefore has limited practical significance.
3 The paper’s tone is conversational (“we first revisit…”, “we then extend…”) and the exposition lacks precision.
Some equations and definitions are introduced loosely, and related work is not carefully positioned against the most recent 2024–2025 literature on LLM inference acceleration.

**Questions:**

The paper does not address when the principle might fail (e.g., under high temperature, small batch size, or hardware constraints).
Practical deployment concerns (GPU memory, caching, latency) are absent.

---

> ### Author Response · Authors · 2025-11-14
>
> Thank you for your comments on our submission. Below are our point-by-point responses to your comments:
>
> >  W1: Contribution and Novelty
>
> Our proposed  principle is not merely a restatement of existing pruning rules, but rather **defines the tree topology with the explicit goal of maximizing speedup**. Unlike previous works that relied on predefined constraints (e.g., maximum depth, width, or total number of nodes) to pursue the maximum acceptable length (e.g., Sequoia, EAGLE-2, SpecInfer), **our method breaks these rigid parameter limitations, automatically identifying the optimal tree structure (insight)**.
>
> Our **proposed principle (new mechanism)** is derived as follows:
>
> 1. Re-deriving the relationship between speedup and draft length from first principles (Equation 1), building on prior theoretical foundations but shifting to a new cost-benefit perspective per node..
> 2. Deriving novel expressions for draft length and time savings (Equations 4 and 8) through this lens.
> 3. Proposing a concise, practical principle (Equations 5 and 9) that directly optimizes for speedup rather than adhering to arbitrary constraints.
>
> Based on this principle, we can directly obtain the theoretically optimal tree. Furthermore, due to the superiority of this principle, we can help current methods shift from maximizing the average acceptance length to maximizing the speedup ratio.
>
> > W2: Assumptions on Acceptance Probability
>
> We acknowledge that acceptance rate estimation is a common requirement for existing tree-based speculative sampling methods (e.g., Sequoia’s acceptance vector, EAGLE-2’s draft model confidence). To address potential errors, we explicitly incorporate **an error term (ε in Equation 3)** into our formulation. We further validated the robustness of this assumption through **Experiment 4.5.1 (Table 4)**, where the theoretically derived optimal threshold differed from the empirically observed one by **only 0.001**. This result demonstrates that our assumption is practically viable.
>
> > W3: Presentation and Related Work
>
> We appreciate your correction on the paper’s tone and exposition. Regarding notation: we adopted the same symbols as Leviathan et al. (ICML 2023 Oral) because this work is a widely cited, foundational reference in speculative decoding. Most subsequent works in the field build on its notation, and the notation we used is the same as that paper for consistency to enhance readability.
>
> [1] Yaniv Leviathan, Matan Kalman, and Yossi Matias. "Fast inference from transformers via speculative decoding." In International Conference on Machine Learning, pp. 19274–19286. PMLR, 2023.
>
> > Response to Questions
>
> Your question about practical deployment limitations is valuable. Our proposed principle serves as a general guiding framework, not an algorithm tailored to specific deployments. Its effectiveness is context-dependent: for instance, insufficient hardware parallelism reduces the speedup of speculative decoding, in turn limiting the utility of our principle.
>
> As shown in **Experiment 4.5.2**, the speedup of speculative sampling decreases when the batch size exceeds 32—our method is no exception. It is worth noting that even under such conditions, our method still provides a higher speedup than EAGLE-2 due to its optimized tree structure. We greatly appreciate your question and will discuss these practical limitations in detail in the revised manuscript.
>
> > Closing Remarks
>
> We wish to express our sincere concerns regarding the current rating, particularly as we have received consistently positive feedback from other reviewers. To elaborate, **Reviewer SzNe** commended the paper for its "**reasonable structure and clear writing**", "**easily integrable method**", and "**comprehensive experiments**". **Reviewer Wruo** further highlighted our work’s standout strengths: "**Principled formulation**", "**Strong empirical validation**", and "**Plug-and-play applicability**"—all core goals we pursued to advance tree-based speculative decoding.
>
> Importantly, these strengths are the product of immense dedication. Every detail of the methodology and experimental design was carefully deliberated, refined, and cross-checked to ensure both theoretical soundness and real-world relevance. We view ICLR as a premier professional platform that fosters constructive academic discussion, and we would deeply value additional feedback to further strengthen our work.

---

### Official Review · Reviewer_Wruo · 2025-10-31

**Soundness:** 3
**Presentation:** 3
**Contribution:** 3
**Rating:** 6
**Confidence:** 5

**Summary:**

This paper proposes a generic tree construction principle (GT principle) for speculative sampling that directly maximizes inference speedup—rather than optimizing the proxy metric of average acceptance length. By redefining per-node cost and benefit and proving the convexity of time reduction with respect to tree size, the method selects only nodes with positive net benefit. Applied to both static (Sequoia) and dynamic (EAGLE-2) tree methods, it achieves consistent end-to-end speedups of 1.97×–2.68× across model sizes.

**Strengths:**

1. Principled formulation: The paper provides a clear, theoretically motivated criterion for tree construction, moving beyond heuristic design. It correctly identifies the core limitation of prior work—optimizing for average acceptance length as a proxy—which can be misaligned with the true objective of minimizing total inference time.
2. Strong empirical validation: Results across model sizes (7B–33B), tasks, temperatures, and batch sizes demonstrate robustness. The paper goes beyond simple benchmarking by including ablation studies on memory efficiency, sensitivity to hyperparameters (like the cost threshold `c`), and generalization across different LLM architectures (Llama-2/3, Qwen2, Vicuna).
3. Plug-and-play applicability: GT integrates seamlessly with both static (Sequoia) and dynamic (EAGLE-2) methods, showing broad utility. The simplicity of the principle—select nodes where estimated benefit outweighs cost—makes it highly practical and easy to adopt by the community.

**Weaknesses:**

1. The GT principle evaluates nodes based on individual net benefit, assuming independence. However, in speculative decoding trees, the utility of a node may stem from its role in enabling high-value subtrees—such as acting as a necessary prefix for multiple high-probability continuations—making isolated evaluation potentially suboptimal.
1. In the integration with EAGLE-2, the adaptive tree depth is governed solely by the acceptance probability along the leftmost path. This choice lacks theoretical justification and may overlook more informative aggregation strategies that better reflect overall tree quality at each depth.
1. The GT principle relies on accurate estimates of token acceptance probabilities to compute node benefit. Since the true acceptance rate ε is unobservable during inference, any systematic bias or variance in its proxy could degrade the effectiveness of node selection.

**Questions:**

1. Could the GT principle miss structurally important “bridge” nodes—those with low immediate benefit but essential for reaching high-benefit descendants—due to its greedy, node-wise selection criterion? If so, how might one extend the framework to account for subtree-level value?
2. Why is the leftmost path used as the sole indicator for adaptive depth in the EAGLE-2 integration? Have alternatives—such as using the maximum, average, or entropy-weighted acceptance probability across all nodes at a given depth—been considered, and how do they affect speedup and stability?
3. In real-world deployment, the acceptance probability must be approximated. How robust is the GT principle to inaccuracies in this estimation? Are there practical strategies (e.g., calibration) to mitigate performance degradation under estimation noise?

---

> ### Author Response · Authors · 2025-11-18
>
> Thank you for your insightful comments. Below is our concise, integrated response addressing each set of related weaknesses and questions:
>
> > W1 and Q1: "Missing Bridge Node" and Subtree Value
>
> A core property of our theoretical framework ensures **a child node’s net benefit cannot exceed its parent’s**. This stems from the child’s estimated acceptance probability being the product of the parent’s acceptance probability and the child’s conditional acceptance rate.
>
> A numerical example illustrates this:  with $M_T=1$ (target model time cost),  $M_D=0.3$ (draft model time cost), $\alpha_0=0.2$ (Node 0’s acceptance probability) and $\alpha_1=0.2*0.99$ (Node 1’s acceptance probability). Therefore, the time reduction can be calculated as follows:
>
> | Node | Benefit      | Cost    | $\Delta T$（Time Reduction） |
> | ---- | ------------ | ------- | ---------------------------- |
> | 0    | 0.2          | 0.3     | 0.2-0.3=-0.1                 |
> | 1    | 0.2+0.2*0.99 | 0.3+0.3 | 0.2+0.2*0.99-0.6=-0.202      |
>
> From the calculations above, even with a high conditional acceptance rate (0.99), the child node’s net benefit does not exceed its parent’s (-0.1 > -0.202). This property eliminates "bridge node" risk: for a descendant to have positive net benefit, its parent must first have a non-negative net benefit (i.e., ≥ 0). Since the GT principle retains parent nodes with non-negative net benefit, this inherent constraint prevents the loss of subtrees that contain nodes with positive net benefit.
>
> > W2 and Q2: The Depth Indicator of EAGLE-2
>
> We appreciate your observation. Our reference to the "leftmost node" per layer was imprecise—**the correct criterion is the highest-probability node within each layer.** Since the probabilities of individual subtrees vary, our method does not prioritize leftmost positions; however, nodes are sorted in descending order of probability from left to right, resulting in a left-skewed tree structure. Concrete examples of this sorting and structural characteristic are provided in **Figures 5 and 6 (Lines 897–935).**
>
> By design, EAGLE-2 generates nodes layer-by-layer rather than individually. To avoid substantial modifications to its original algorithmic and code logic, we only enforce one key constraint: that at least one node in the layer satisfies our GT principle. We check the highest-probability node to confirm layer validity, and all nodes in the layer that meet the principle are subsequently selected.
>
> > W3 and Q3: Error term of Acceptance Probability
>
> To address the unobservability of the true acceptance probability and potential estimation biases, we explicitly incorporate an **error term (ε, Equation 3)** into our formulations—consistently included in our core GT principles (Equations 5 and 9). Validated via **Experiment 4.5.1 (Table 4)**, a 0.001 discrepancy between theoretical and empirical optimal thresholds confirms negligible estimation noise impact, directly validating the GT principle’s robustness to systematic biases and random variance in estimation.
>
>
> Thank you for your valuable review and insights. We hope our responses address your questions adequately. Please feel free to reach out with further inquiries or for additional discussions—we are happy to provide supplements as needed. Thank you again for your contributions.

---

### Official Review · Reviewer_AyBk · 2025-11-03

**Soundness:** 2
**Presentation:** 3
**Contribution:** 2
**Rating:** 2
**Confidence:** 4

**Summary:**

- The paper proposes an early stopping algorithm for tree-based speculative decoding to reduce extra draft runs, aiming to improve speed-up with only a slight reduction in acceptance.
- Claims that expected speed-up is a convex function over draft length—but this seems like a known property already.

**Strengths:**

- Introduces a practical idea to reduce unnecessary draft generation, which can save compute during inference.
- Focuses on speed-up optimization, which is critical for real-world deployment of speculative decoding.

**Weaknesses:**

- No comparison with adaptive draft length methods, despite the proposed approach being conceptually similar (stopping draft generation dynamically).
- Eq. (4) is missing $M_T$​ on $\epsilon$; text below Eq. (4) should include $M_T \epsilon$ since they come together.
- Full form of GT is unclear—should clarify if it means General Tree.
- Criticism of Sequoia in the static tree section seems invalid:
  - Sequoia is offline, so its cost is a one-time calibration, which is standard and not a runtime overhead unless extremely large.
  - The claim that Sequoia fails to find an optimal speed-up tree needs evidence.
- Eq. (10) and Eq. (11) lack intuition—currently appear arbitrary.
- Improvement in speed for static tree in Table 1 is marginal (second decimal point).
- Fig. 3 legend says “GE,” likely should be “GT.”
- Table 4 does not specify the evaluation task.
- Overall, the paper does not convincingly show significant gains compared to existing methods.

**Questions:**

- Please check weaknesses
- Can the authors include SpecBench comparisons using EAGLE2/3 with LLaMA3 family, I'm curious to see performance on models with  large vocabulary models?

---

> ### Author Response · Authors · 2025-11-17
> **Response to Reviewer AyBk - Part 1**
>
> Thank you sincerely for your meticulous review and valuable comments on our submission. We greatly appreciate the insights you have provided, which have helped us clarify key points and improve our work. Below is our detailed response to each comment. Due to word limits, the response is divided into two parts:
>
> > Response to Summary
>
> First, we would like to clarify a potential misunderstanding—likely arising from insufficient clarity in our original expression: **Our contribution is not merely an early stopping algorithm, but a tree topology construction principle designed to directly maximize speedup.** Previous works, such as Sequoia and EAGLE-2, are bound by predefined constraints (e.g., max depth, width, or node count) in their pursuit of acceptance length. In contrast, our method bypasses these rigid parameters, offering the insight to automatically determine the optimal tree structure derived from first principles.
>
> Regarding the convexity of expected speed-up with respect to draft length: while this phenomenon is intuitively plausible, prior works have rarely provided rigorous analysis or derivations to formalize this property **(see Eq. (1), Lines 142–145)**. In contrast, our newly proposed formulations **(Eq. (4) and (8), Lines 182–184 and 207–209)** directly capture this convexity.
>
> > W1: Comparison with adaptive draft length methods
>
> We have conducted comparative experiments with **EAGLE2, the state-of-the-art adaptive tree method**. A key limitation of EAGLE2 is that its dynamic draft length is constrained by a pre-set maximum length parameter. Our GT principle breaks this constraint by dynamically determining both the draft length upper bound and optimal tree structure, rather than being confined to pre-defined parameters. Detailed results are presented in **Table 1 (Lines 270–294)**.
>
> > W2: Missing term in Eq. (4)
>
> Thank you for your careful check of the equations. We confirm that the entire term $M_T e$ in Eq. (4) is an error term, and we use $e$ to denote it specifically to enhance the manuscript's readability.
>
> > W3: Clarification of "GT"
>
> We apologize for the insufficient clarity. **"GT" stands for "Greater Than"**, the core logic of our proposed principle, which is described in **Lines 194 and 215 of Page 4**. We will add this definition explicitly at the first occurrence of "GT" in the revised manuscript.
>
> > W4: Criticism of Sequoia
>
> We would like to clarify **the "cost" we referenced**: it does not refer to offline tree construction cost (which we acknowledge is a standard one-time calibration), but rather **runtime draft generation cost**—i.e., the computational overhead when the draft model generates tokens based on the pre-constructed tree. Similar to EAGLE2, Sequoia relies on pre-set maximum depth and node selection criteria, so its optimized tree only maximizes average acceptance length under these constraints. Our GT principle breaks these pre-defined limits and directly optimizes for maximum speed-up ratio, rather than suboptimal targets under constraints.
>
> > W5: Intuition for Eq. (10) and Eq. (11)
>
> Thank you for your comment on the Equations (10) and (11). We use Sequoia—an approach that pre-computes the acceptance vector for tree construction—as a concrete illustrative example to demonstrate how our proposed GT principle is applied. The core logic of these two equations is to **incorporate all nodes that satisfy the GT principle into the tree structure**; this ensures the constructed tree can achieve the optimal speed-up ratio. Specifically, the tree depth is determined by the greatest acceptance vector: given the condition $a_{max}^d > c$, the depth $d$ can be derived as $d = \log_a c$. Meanwhile, the total number of nodes in the tree is exactly the count of all nodes that meet the GT principle.
>
> > W6: Marginal improvement in static tree (Table 1)
>
> The baseline methods in Table 1 deliver solid performance thanks to careful parameter tuning. However, this parameter-tuning approach lacks robustness: **when a new model is adopted, pre-tuned parameters become invalid and necessitate time-consuming re-optimization.** In contrast, our GT principle requires no parameter tuning at all and can be directly applied to new models. For instance, parameter settings optimized for EAGLE2 are incompatible with EAGLE3(**Table 6, line 756-784**);, whereas our principle automatically identifies the optimal draft length—yielding both higher speed-up gains and longer average acceptance lengths. Furthermore, we have validated the GT principle on various other static tree structures (**including MEDUSA, EAGLE, and HYDRA, as presented in Table 6**), where our method consistently achieves significant performance improvements. The results in Table 1 are thus intended to fairly demonstrate that our method not only matches—if not surpasses—the performance of manually tuned baselines, but also offers notable advantages in generalization and robustness.

---

> ### Author Response · Authors · 2025-11-17
> **Response to Reviewer AyBk - Part 2**
>
> Due to character limits, the reply is divided into two parts. The following is the second part:
>
>
>
> > W7: "GE" in Fig. 3 legend
>
> Thank you for pointing out this typo. We will correct "GE" to "GT" in the revised manuscript.
>
> > W8: Unspecified evaluation task in Table 4
>
> The evaluation task for Table 4 is **mt-bench**, which is mentioned in **Line 377 of Page 7.** We will add a clear note on the evaluation task directly in the caption of Table 4 in the revised manuscript for better readability.
>
> > W9: Lack of convincing significant gains
>
> As a general optimization principle, our method offers two key advantages: (1) **Performance superiority**: It outperforms baselines with well-tuned parameters (e.g., 0.40× speed-up improvement for LLaMA3, shown in Table 2); (2) **Strong adaptability**: It is compatible with various new models and methods. For example, when applied to EAGLE-3 (NeurIPS 2025) on Vicuna-13B with the mt-bench task, it achieves a 2.40 improvement in average acceptance length and a 0.27× speed-up gain (Table 6). This adaptability enables our principle to support the optimization of future draft models and tree structures, which we believe constitutes a significant contribution beyond marginal performance gains.
>
> [1] Yuhui Li, Fangyun Wei, Chao Zhang, and Hongyang Zhang. EAGLE-3: Scaling up inference acceleration of large language models via training-time test. In Annual Conference on Neural Information Processing Systems, 2025.
>
> > Response to Questions
>
> Regarding the comparison on SpecBench with EAGLE2/3 and the LLaMA3 family: We have conducted relevant experiments with **EAGLE2** using **LLaMA3** as the base model on mt-bench, and the results are presented in **Tables 2 and 5**.
>
> Once again, we thank you for your constructive comments. We believe these revisions will significantly improve the quality and clarity of our manuscript. Please feel free to let us know if you have any further questions or suggestions.

---

### Official Review · Reviewer_SzNe · 2025-11-03

**Soundness:** 3
**Presentation:** 4
**Contribution:** 3
**Rating:** 6
**Confidence:** 2

**Summary:**

This paper proposes a "Generic Tree Construction Principle" (GT principle) for tree-based speculative sampling in large language models (LLMs), aiming to maximize inference speedup by selecting nodes with positive net benefits based on redefined costs and benefits. The authors derive a convex function for time reduction, extend it to tree structures, and integrate it into existing methods like Sequoia and EAGLE-2, claiming 4-14% performance gains and speedups up to 2.68×. Experiments are conducted on benchmarks like Spec-Bench with models from 7B to 33B parameters.

**Strengths:**

1. This paper has a reasonable structure and clear writing. The authors elaborate on the research methods in detail, making it highly readable.
2. The proposed method can be easily integrated with existing speculative decoding algorithms to achieve performance improvements.
3. The experiments are comprehensive, covering multiple model scales (Vicuna with 7B–33B parameters), tasks (machine translation, transformation, summarization, question answering, multi-turn dialogue, and retrieval-augmented generation from Spec-Bench), and baseline methods (Sequoia, EAGLE-2, with additional methods including EAGLE, EAGLE-3, Medusa, and Hydra in the appendix).

**Weaknesses:**

1. Although the method proposed by the authors achieves a 4%-14% performance improvement compared to the baselines, the improvement margin is relatively small. The end-to-end speedup is largely contributed by the baseline methods, and the enhancement brought by the authors' method to the baselines is quite limited. Additionally, the improvement in the R value (memory-to-speedup ratio) is negligible.
2. Code generation tasks (such as the HumanEval benchmark) are not included, and there are few experimental results on larger models (e.g., 70B models).

**Questions:**

1. Could you provide the quality metrics for each benchmark in Spec-Bench (such as BLEU for the WMT task, accuracy for the GSM8K task, etc.) to demonstrate that the GT principle does not reduce the utility of the model?
2. The authors claim that "all experiments were repeated 5 times under the same hyperparameters, and the final results are presented as averages". Could you please provide the variance of the 5 experiments?

---

> ### Author Response · Authors · 2025-11-18
>
> Thank you for your valuable feedback and thoughtful evaluation of our submission. We appreciate your recognition of the paper’s structure, readability, and comprehensive experiments, as well as your constructive comments on its weaknesses and questions. Below is our detailed response to address your concerns:
>
> > W1: Regarding the "relatively small improvement margin" and "limited enhancement to baselines
>
> Our work proposes a **general optimization principle** (GT principle) rather than a new model. The baseline methods (e.g., Sequoia, EAGLE-2) have already been well-optimized, making further improvements inherently challenging. Notably, the GT principle exhibits **strong adaptability to emerging draft models**—for EAGLE-3 (NeurIPS 2025), the dynamic strategy originally proposed by EAGLE-2 can no longer keep up with its draft model advancements. The results in Table 6 (lines 781-784) show that integrating our GT Principle significantly improves the average draft length and speedup of EAGLE-3, demonstrating the principle's practical value in evolutionary speculative decoding frameworks.
>
> [1] Yuhui Li, Fangyun Wei, Chao Zhang, and Hongyang Zhang. EAGLE-3: Scaling up inference acceleration of large language models via training-time test. In Annual Conference on Neural Information Processing Systems, 2025.
>
> > W2: Regarding the absence of code generation tasks (e.g., HumanEval) and larger-model experiments (e.g., 70B):
>
> To ensure fair comparison with existing speculative decoding works, we strictly adhered to the Spec-Bench dataset—the standard benchmark for this research field. Notably, our GT principle **still exhibits strong performance on code generation tasks.** To further validate its effectiveness, we include comparative results between the GT principle and EAGLE-3 on the humaneval dataset in the table below. Additionally, experimental results for the 70B model are detailed in **Section 4.5.2 (lines 393-412).**
>
>
>
> |              | Speedup   | Average Acceptance Length |
> | ------------ | --------- | ----------------------- |
> | EAGLE3       | 3.93x     | 6.50                    |
> | GT           | 4.73x     | 10.59                   |
> | $\Delta Imp$ | **0.80x** | **4.09**                |
>
> The results indicate that the GT principle delivers a **0.80×** speedup enhancement and a **4.09** increase in average acceptance length. These notable gains further underscore the strong adaptability and effectiveness of our method.
>
> > Q1: Quality metrics for Spec-Bench benchmarks:
>
> To clarify, we wonder if your question centers on whether our method impacts the model’s output? If we have misinterpreted your query, we apologize sincerely. Speculative decoding requires that the final generated results are **identical to those of the original decoding method** (i.e., no deviation from the target model’s outputs). This core constraint means quality metrics are not applicable—consistency with the original outputs is the primary quality guarantee, which our method strictly adheres to. However, if the intention is to assess the generation quality of the draft model specifically, we consider the **average acceptance length ($L$)** to be a robust and appropriate metric (**Table 1**).
>
> > Q2: Variance of the 5 repeated experiments:
>
> Owing to space constraints, we did not incorporate the variance data in the initial submission. To respond to your question, we present the variance data of speedup for the GT principle and EAGLE-2 on Spec-Bench datasets using Vicuna-7b below, aiming to further enhance the transparency of our statistical findings.
>
> | Method | MT     | Trans  | Sum    | QA     | MR     | RAG    | Overall |
> | ------ | ------ | ------ | ------ | ------ | ------ | ------ | ------- |
> | EAGLE2 | 0.0060 | 0.0022 | 0.0032 | 0.0042 | 0.0085 | 0.0022 | 0.0041  |
> | GT     | 0.0022 | 0.0007 | 0.0005 | 0.0005 | 0.0002 | 0.0001 | 0.0003  |
>
> Our findings indicate that the observed variance (0.0041, 0.0003) is negligibly small, thereby confirming the reliability of our results.
>
> We sincerely thank you for your valuable and insightful comments. We warmly welcome any further discussions or additional clarifications you may require.

---

### Note · Authors · 2025-12-03

I have read and agree with the venue's withdrawal policy on behalf of myself and my co-authors.